



# Reviews and syntheses: Processes and functional genes involved in nitrogen cycling in marine environments

Ramiro Ramos[1,2], Silvia Pajares[2]

[1]Posgrado en Ciencias del Mar y Limnología, Universidad Nacional Autónoma de México, Mexico City, 045110, Mexico
[2]Unidad Académica de Ecología y Biodiversidad Acuática, Instituto de Ciencias del Mar y Limnología, Universidad Nacional Autónoma de México, Mexico City, 045110, Mexico

*Correspondence to*: Silvia Pajares (spajares@cmarl.unam.mx)

**Abstract.** Nitrogen is a key element for life in the oceans. It controls primary productivity in many parts of the global ocean, consequently playing a crucial role in the uptake of atmospheric carbon dioxide. The nitrogen cycle is driven by complex
biogeochemical transformations mediated by microorganisms, including classical processes such as nitrogen fixation, assimilation, nitrification, denitrification, and dissimilarity nitrate reduction to ammonia, as well as novel processes such as anaerobic ammonium oxidation, comammox and nitrite-driven anaerobic methane oxidation. The nitrogen cycle maintains the functioning of marine ecosystems and will be a crucial component in how the ocean responds to global environmental change. In this review, we summarize the current understanding of the marine microbial nitrogen cycle, its underlying
biochemical and enzymatic reactions, the ecology and distribution of the microorganisms involved, and the main impacts of anthropogenic activities.

## 1    Introduction

Nitrogen (N) is a key element for life and the functioning of marine ecosystems. It plays a crucial role in marine biogeochemistry, and because of its connections to the cycles of other elements such as carbon (C) it has a strong impact on
Earth's climate (Gruber, 2008). N is scarce in the ocean, and consequently limits marine productivity in many parts of that biome (Glibert et al., 2016). Its availability also regulates the strength of the biological pump, one of the mechanisms contributing to oceanic uptake of C dioxide ($CO_2$) (Falkowski, 1997). Furthermore, one of its forms, nitrous oxide ($N_2O$), is a powerful greenhouse gas and an ozone-depleting agent.

N is quite a versatile element and in the ocean is present in different oxidation states, ranging from -III in reduced forms like
ammonium ($NH_4^+$) and organic N to +V in fully oxidized nitrate ($NO_3^-$), which highlights its importance as both an electron acceptor and donor for energy metabolism in marine ecosystems (Fig. 1). Microorganisms mainly mediate the redox transformations of N, changing the concentrations of N compounds in the environment (Kuypers et al., 2018). Thus, microbial communities related to marine N cycling have been studied extensively using both culture-dependent and independent techniques (Guerinot and Colwell, 1985; Rasigraf et al., 2017). Technological advances in nucleic acid
sequencing have allowed the attainment of numerous genomic data over the past two decades, revealing enormous metabolic



versatility within N-transforming microorganisms (Kuypers et al., 2018). Furthermore, the study of genes encoding key metabolic proteins has provided important discoveries about the biological potential of microorganisms participating in N cycling processes and has given deep insight into the ecological factors that determine their biogeography and activity in marine systems (Damashek and Francis, 2017; Devol, 2005; Lam and Kuypers, 2011).

The vast majority of N exists as atmospheric dinitrogen gas ($N_2$), which is only available to $N_2$-fixing bacteria and archaea. Thus the major sources of fixed N for the ocean are biological $N_2$ fixation and atmospheric deposition, while the major sinks are denitrification and anammox. Because alterations of this balance caused by anthropogenic activity may pose significant impact on marine ecosystem health, biodiversity and climate change, the study of microbial communities involved in marine N cycling has gained great interest in recent years (Landolfi et al., 2017; Voss et al., 2013). To predict this impact, there is an

urgent need to understand the basic mechanisms that underlie microbial N processes in marine ecosystems.

As a contribution to this understanding, the present review provides a general survey of the microbial processes that comprise the marine N cycle. This includes novel processes and addresses the biochemical and enzymatic reactions involved, the genetic machinery that makes it possible, and the ecology and distribution of participating microorganisms in different marine ecosystems. We also identify several of the knowledge gaps that we still face in the study of microbial

marine N processes. We end our review with a discussion of the impacts that anthropogenic activity has on the microbially mediated mechanisms regulating the marine N cycle.

## 2    Microbial processes in the marine N cycle

The N cycle is driven by microbial transformations, including classical processes such as $N_2$ fixation, assimilation, nitrification, denitrification, and dissimilarity nitrate reduction to ammonia, as well as novel processes such as anaerobic

ammonium oxidation, complete ammonia oxidation (comammox) to nitrate, and nitrate/nitrite-dependent anaerobic methane oxidation, which complicates the already intricate picture of marine N cycling. N-converting enzymes are found in very diverse microorganisms that are globally distributed throughout marine systems, from estuarine and coastal zones to open oceans, including oxygen minimum zones (OMZs) and deep-sea environments (Fig. 2); this emphasizes their key environmental function in such systems.  In addition, novel ocean microorganisms have been identified in recent years, such

as symbiotic heterotrophic $N_2$-fixing cyanobacteria (Thompson et al., 2012), ammonia-oxidizing archaea (Konneke et al., 2005), and denitrifying eukaryotic foraminifera (Risgaard-Petersen et al., 2006).

### 2.1    $N_2$ fixation

Marine $N_2$ fixers are able to convert dissolved $N_2$ gas into bioavailable N for planktonic production, according to the following reaction:

$N_2 + 8H^+ + 8e^- + 16ATP \rightarrow 2NH_3 + H_2 + 16ADP + 16\ P_i$



N$_2$ is a very stable molecule, thus its fixation is an intensely energy-requiring process that only one group of microorganisms called diazotrophs is able to achieve. Marine diazotrophs mainly include cyanobacteria such as non-heterocystous filamentous cyanobacteria (e.g., *Trichodesmium, Oscillatoria* and *Lyngbya*), heterocystous filamentous cyanobacteria (e.g., *Aphanizomenon* and *Nodularia*), unicellular cyanobacteria (UCYN, such as the uncultivated Group A and *Crocosphaera*

*watsonii* [Group B]), and diatom symbiotic cyanobacteria (such as *Richelia intracellularis*). Other marine diazotrophs include heterotrophic bacteria (e.g., *Klebsiella, Vibrio* and *Azotobacter*), phototrophic bacteria (such as *Chlorobium, Chromatium* and *Rhodospirillum*), strict anaerobes (e.g., *Clostridium* and *Desulfovibrio*), iron (Fe) oxidizers (e.g., *Thiobacillus*) and methanogenic Euryarchaeota (Karl et al., 2002; Zehr and Paerl, 2008). These microorganisms are quite diverse but share a common feature: the nitrogenase complex, which catalyzes N$_2$ fixation.

The nitrogenase complex is composed of two multisubunit metalloproteins: the molybdenum (Mo)–Fe protein (Component I) encoded in the *nifD* and *nifK* genes; and the Fe protein (Component II) encoded in the *nifH* gene. Alternative nitrogenases replace Mo with vanadium (V) or solely contain Fe in Component I, encoded in the *vnfH* and *anfH* genes, respectively (Zehr et al., 2003). Nitrogenase is the preferred functional biomarker in the study of diazotroph abundance and diversity because it remains highly conserved (Table S1 in the Supplement). Marine environmental *nifH* sequences group into four major

clusters designated I–IV (Jayakumar et al., 2012; Turk et al., 2011; Zehr et al., 2003), although three newly putative clusters (clusters III-x, V, and VI) have recently been identified (Dang et al., 2009b).

### 2.1.1 Factors affecting N$_2$ fixation in marine systems

Light, oxygen (O$_2$), temperature, trace metals, sulfate (SO$_4^{-2}$), and several forms of inorganic N are among the main factors that affect marine diazotroph community composition. It is well known that the nitrogenase complex is inactivated by O$_2$ and

its reactive species, and diazotrophs have developed numerous protective mechanisms against this. For example, several cyanobacteria generate specialized N$_2$-fixing cells called heterocysts that provide an almost anoxic environment. Temporal separation is another protection mechanism; most photosynthetic diazotrophs fix N$_2$ at night when O$_2$ is not being produced, although a number of exceptions have been identified. Zehr et al. (2008) reported the discovery of a *Cyanothece*-related cyanobacterium that does not possess photosystem II genes and, therefore, is able to fix N$_2$ during both day and night.

Another mechanism was found in *Trichodesmium*, which develops diazocytes — specialized cells that share several characteristics with heterocysts — and fixes N$_2$ during the photoperiod while simultaneously producing O$_2$ (Berman-Frank et al., 2001b). The coupling of these processes in *Trichodesmium* makes light another regulating factor for N$_2$ fixation. Capone et al. (1990) suggest that peak activity occurs around midday, hence light may be a determinant factor for N$_2$ fixation.

N$_2$ fixation is not intrinsically restricted by temperature and can occur near 0 °C (Stal, 2009), although temperature does

seem to set the limits of where different diazotrophs can exist. The distribution of *Trichodesmium*, the most commonly studied diazotroph, appears to be well constrained to warmer tropical and subtropical surface waters (Breitbarth et al., 2007; Capone et al., 1997), while small diazotrophs have been mainly found in colder surface waters (Holl et al., 2007; Needoba et



al., 2007). However, the presence of unicellular diazotrophs has also been reported in subtropical and tropical waters of the Atlantic Ocean (Foster et al., 2009; Montoya et al., 2009) and Pacific Ocean (Bonnet et al., 2009), as well as in the Arabian Sea (Mazard et al., 2004). It is important to note that temperature may be correlated with other factors that control the distribution patterns of marine diazotrophs such as $NO_3^-$ or $O_2$ (Sohm et al., 2011).

It is generally assumed that inorganic N is able to suppress $N_2$ fixation in cyanobacterial diazotrophs due to the additional energetic cost associated with assimilating $N_2$ relative to $NH_4^+$ or $NO_3^-$ (Knapp, 2012), although diazotrophs can have different responses to these inorganic N compounds. For instance, it has been shown that $NO_3^-$ and $NH_4^+$ inhibit $N_2$ fixation in *Trichodesmium* (Holl et al., 2005) and *Nodularia* (Vintila and El-Shehawy, 2007), respectively, but not in *Anabaena* (Ohmori and Hattori, 1974).

Fe and Mo are enzymatic cofactors, and their bioavailability directly affects $N_2$ fixation in certain areas of the ocean (Berman-Frank et al., 2001a; Karl et al., 2002). Fe is generally depleted in surface waters of the open ocean, and the delivery of dust rich in Fe to the ocean may ultimately control the rate of $N_2$ fixation on the global ocean scale (Sohm et al., 2011). For instance, *Trichodesmium* is abundant in the North Atlantic Ocean, in which dissolved Fe concentrations are relatively high because dust inputs are greater than in the South Atlantic Ocean, where dissolved Fe concentrations are extremely low

(Moore et al., 2009). In addition, $SO_4^{-2}$ indirectly inhibits $N_2$ fixation because it reacts with Mo to form a structural analog, molybdate ($MoO_4^{2-}$), which competes with Mo and impedes $N_2$ fixation (Karl et al., 2002; Zehr and Paerl, 2008).

### 2.1.2 Distribution of diazotrophs in marine environments

$N_2$ fixation seems to be an important process, not only in tropical surface waters (Capone et al., 2005) but also in hypoxic waters (Hamersley et al., 2011), deep sea (Dekas et al., 2009), hydrothermal vents (Mehta and Baross, 2006) and coral reefs

(Lema et al., 2012), as well as in estuaries and nutrient-rich coastal upwelling regions (Wen et al., 2017). The main findings concerning the distribution of diazotrophs in several marine environments are summarized below.

*Open oceans*

Due to the energetic demand that it poses, $N_2$-fixation is a clear advantage in extremely oligotrophic environments such as

ocean gyres, in which reactive N (Nr) is scarce (Capone et al., 2005; Karl et al., 2002). The abundance and distribution of the *nifH* gene have been studied in different areas of the open ocean; much of this research has focused on *Trichodesmium,* which has a cosmopolitan distribution throughout the nutrient-poor tropical and subtropical seas where it often forms massive surface blooms (Capone et al., 2005). Although the three main diazotroph types (*Trichodesmium* analogs, UCYN, and diatom diazotroph associations) seem to coexist in the ocean, *Trichodesmium* dominates in the North and Tropical

Atlantic Ocean and the Arabian Sea, the UCYN — particularly Group A — dominates in the South Atlantic, Pacific and Indian oceans, and diatom diazotroph associations are widely distributed through the warm oligotrophic ocean, with the largest densities in the Amazon River plume (Monteiro et al., 2010; Sohm et al., 2011).



The dominance of *Trichodesmium* in the warmer waters of the Tropical and Subtropical Atlantic Ocean seems to be due to low concentrations of inorganic N forms (Goebel et al., 2010; Langlois et al., 2005), while its dominance in the Northern Atlantic Ocean seems to be due to the higher and lower concentrations of dissolved Fe and inorganic P, respectively (Langlois et al., 2008; Moore et al., 2009). The most abundant diazotrophs in the Eastern Equatorial Atlantic appear to be

*Trichodesmium* and UCYN Group A, and their distributions are controlled by riverine inputs and upwelling (Foster et al., 2009). In the Pacific Ocean, UCYN groups are mainly restricted to the well-lit, nutrient-poor waters of the North Pacific Subtropical Gyre (Church et al., 2008; Gradoville et al., 2017; Needoba et al., 2007), while heterotrophs (mainly Alpha- and Betaproteobacteria) dominate the diazotrophic community in the Eastern South Pacific and Pacific Northwest coastal upwelling systems (Gradoville et al., 2017). In general, diazotrophs are in lower abundances in the South Pacific Ocean, and

temperature and depth seem to control the distribution of the two UCYN groups: *Crocosphaera* dominates in warmer surface waters, while Group A does so in cooler waters (Moisander et al., 2010). In addition, high abundances of Group A can be found at higher latitudes and deeper waters than *Trichodesmium* in the South Pacific Ocean (Moisander et al., 2010). $N_2$ fixation in the Indian Ocean is poorly understood and seems to be mainly performed by heterotrophic bacteria (Shiozaki et al., 2014), except in the Arabian Sea where persistent *Trichodesmium* blooms occur (Capone et al., 1998).

*Estuaries and coastal zones*

Little is known about the distribution and activity of diazotrophs in estuaries and coastal regions. $N_2$ fixation in these systems is affected by high nutrient inputs from land, so it is assumed to be unimportant (Howarth et al., 1988). However, recent studies suggest that $N_2$ fixation may be an important source of Nr in a number of estuaries and nutrient-repleted coastal

upwelling regions (Fulweiler et al., 2007; Mulholland et al., 2012; Wen et al., 2017). Also, studies of diversity, abundance, and expression of the *nifH* gene have suggested complicated relationships between environmental drivers and diazotroph distribution in these ecosystems (Moisander et al., 2007; Severin et al., 2015; Short et al., 2004). In the Baltic Sea, for instance, planktonic heterotrophic diazotrophs were found to be common in a eutrophic estuary while cyanobacterial diazotrophs were more abundant in a lower-nutrient estuary (Bentzon-Tilia et al., 2015). In the coastal waters of the Mid-

Atlantic continental shelf, UCYN-A was the most abundant diazotroph (Mulholland et al., 2012), while diatom symbiotic cyanobacteria dominated over the other diazotrophic groups in the upwelling regions of the Taiwan Strait where high diazotrophic activity was present (Wen et al., 2017).

*Deep-sea environments*

Mehta *et al.* (2003) provided the first evidence of *nifH* genes, mostly belonging to clusters II and III, in deep-sea hydrothermal vent fluid. A few years later Mehta and Baross (2006) isolated a methanogenic archaeon from said environment that was able to fix $N_2$ at 92°C. Potential $N_2$ fixation by methanogenic archaea was also found in seep sediments from the Kumano Basin in Japan (Miyazaki et al., 2009). Cao et al. (2015) compared the abundance and diversity of *nifH* sequences in hydrothermal vents from four middle ocean ridges; being cluster I, comprised by proteobacterial and





cyanobacterial *nifH* sequences, the most dominant group. Wu et al. (2014) found that *nifH* sequences belonging to proteobacterial clusters I and III dominate $N_2$ fixation in the sediments from a deep-sea hydrothermal vent in the Southwest Indian Ridge. In addition, *nifH* sequences retrieved from methane seep sediments in the Okhotsk Sea were placed into three new putative clusters (III-x, V and VI). Sequences from cluster V were unique to these sediments, and sequences from

cluster III-x are unique to the deep-sea methane seep environments studied to date (Dang et al., 2009b).

An intriguing microbial consortium was isolated from a methane cold seep in Eel River Basin, California (Dekas et al., 2009). The consortium, composed of an anaerobic methanotrophic archaeon (ANME) and a sulfate-reducing bacterium (SRB), mediated both sulfate-dependent anaerobic methane oxidation (SDAMO) and $N_2$ fixation. This is interesting from an energetic point of view, given that $N_2$ fixation is a very energy-demanding process and SDAMO is one of the least

energetically productive metabolisms (Strous and Jetten, 2004). In addition, a study conducted in methane seep sediments in Costa Rica showed that the distribution of $N_2$ fixation by this consortium is heterogeneous and is likely influenced by chemical gradients (Dekas et al., 2014).

*Coral reefs*

Corals are found in N-depleted tropical and subtropical coastal waters; thus, $N_2$ fixation should play an important role in the ecosystems there, providing an additional source of N for symbiotic dinoflagellates and thus improving the productivity of those systems (Lema et al., 2012). In fact, several studies have found evidence of $N_2$-fixing symbionts in coral reefs, heterotrophic Proteobacteria being the dominant diazotrophs in different types of coral holobionts (Lema et al., 2012; Lema et al., 2014; Olson, 2009; Olson and Lesser, 2013). Furthermore, diazotrophic communities seem to differ among the trophic

functional groups of coral, suggesting that the ecological importance of $N_2$-fixing symbionts may be determined by the trophic functional group of the host. For instance, Pogoreutz et al. (2017) found that autotrophic Pocilloporidae exhibited *nifH* copies and gene expression 100 times higher than those in heterotrophic Fungiidae, suggesting that $N_2$ fixation compensates for the low heterotrophic N uptake in autotrophic corals.

### 2.2   Nitrification

Nitrification involves three types of microorganisms: 1) those who oxide ammonia ($NH_3$) to nitrite ($NO_2^-$) (nitritation), 2) those who oxide $NO_2^-$ to $NO_3^-$ (nitratation), and 3) those who oxide $NH_4^+$ directly to $NO_3^-$ (comammox, complete ammonium oxidation), which will be addressed in the next section.

(1) $NH_3 + O_2 \rightarrow NO_2^- + 3H^+ + 2e^-$

(2) $2NO_2^- + O_2 \rightarrow 2NO_3^-$

(3) $NH_4^+ + 2O_2 \rightarrow NO_3^- + H_2O + 2H^+$

Reaction (1) is carried out by ammonia oxidizers, which are chemolithotrophs classified in two groups: ammonia-oxidizing bacteria (AOB) belonging to a few genera within the Betaproteobacteria (*Nitrosomonas* and *Nitrosospira*) and



Gammaproteobacteria (*Nitrosococcus*) classes (Purkhold et al., 2000), and ammonia-oxidizing archaea (AOA) belonging to the Thaumarchaeota phylum, such as *Nitrosopumilus maritimus* (Konneke et al., 2005). The discovery of AOA solved the long-standing mystery of the apparently rare ammonia oxidizers in the ocean (Wuchter et al., 2006). This reaction is a two-step process: the first and rate-limiting step is the conversion of $NH_3$ to hydroxylamine ($NH_2OH$) catalyzed by the ammonia

monooxygenase (AMO). The second step was once believed to be the conversion of $NH_2OH$ to $NO_2^-$ by the hydroxylamine oxidoreductase (HAO), but recent studies indicate that $NH_2OH$ is first converted to nitric oxide (NO), then NO is oxidized to $NO_2^-$ under aerobic conditions (Caranto and Lancaster, 2017). This is interesting since Ward (2008) reported the production of NO, $NO_2$ and even $N_2$ from $NH_4^+$ in marine environments, which is enhanced in suboxic conditions.

The AMO enzyme is composed of three subunits: AmoA, AmoB and AmoC. The *amoA* gene, coding for the AmoA, has

been widely used as a molecular marker for studying ammonia oxidizers in the environment (Rotthauwe & Witzel, 1997) (Table S2 in the Supplement). Several considerations must be taken for the analysis of *amoA*-containing microorganisms: AOB belonging to Betaproteobacteria usually possess more than one copy of the gene, while Gammaproteobacteria only possess one copy. Additionally, some primers used in the amplification of this gene can also amplify the *pmoA* genes from methane-oxidizing bacteria, since both genes are phylogenetically related (Norton et al., 2002). The HAO enzyme is a

homotrimer and is encoded in the *hao* gene, present only in AOB with multiple copies (Arp et al., 2007). The *hao* gene has been little-used as a molecular marker of AOB in marine systems (Lüke et al., 2016; Rasigraf et al., 2017).

Reaction (2) is simpler, since it only requires the transference of two electrons and does not produce intermediaries. It is catalyzed by the nitrite oxidoreductase (NXR), which is present in nitrite-oxidizing bacteria (NOB) such as *Chloroflexi*, *Nitrospirae*, *Nitrospinae* and several classes of Proteobacteria (Ward, 2008). Three subunits comprise the NXR enzyme:

NxrA, NxrB and NxrC. The gene encoding NrxA has been used as a marker for studying NOB in a number of marine environments (Lüke et al., 2016; Rani et al., 2017; Rasigraf et al., 2017) (Table S2 in the Supplement).

### 2.2.1    Factors affecting nitrification in marine systems

Oxygen, light, pH, salinity, $NH_4^+$ and $NO_2^-$ are among the main factors controlling nitrification. Nitrifiers require oxygen for metabolism, but they seem to do well in microaerophilic conditions. It has been demonstrated that low $O_2$ and $NO_2^-$

concentrations are correlated with the abundance of AOA in a number of marine environments (Cao et al., 2012; De Corte et al., 2009; Urakawa et al., 2014). Nitrifying organisms avoid light because it affects the cytochromes involved in electron transport; thus, the highest rates of ammonia oxidation often occur just below the sunlit ocean surface (Yool et al., 2007). Additionally, *amoA* transcripts have been reported to be more abundant in the surface of the Arctic Ocean during the darkest winter month as well as in the halocline, where light levels are perpetually lower (Pedneault et al., 2014). Nitrification seems

to be controlled by $NH_4^+$ limitation and, just like any other process involving hydrogen ions ($H^+$), is affected by pH (Ward, 2008, 2013). Temperature does not affect nitrification directly, but can affect community composition in some environments. For example, the composition and abundance of AOA communities in the San Francisco Bay estuary seem to be highly





correlated with temperature (Mosier and Francis, 2008), while marine AOA seem unaffected by it (Horak et al., 2013). Temperature also seems to be an important factor influencing the distribution and diversity of AOA and AOB in sponge associations (Cardoso et al., 2013). Other factors affecting marine nitrifier communities are site-specific. For example, in deep-sea hydrothermal vents, total N and S seem to control the distribution of ammonia oxidizers (Xu et al., 2014).

Moreover, salinity is an important factor driving *amoA* gene diversity in water columns and sediments from estuaries (e.g., Beman and Francis 2006; Bernhard et al., 2005; Bernhard et al., 2010; Bouskill et al., 2011; Dang et al., 2008; Francis et al., 2003; Mosier and Francis, 2008; Smith et al., 2015b).

### 2.2.2   Distribution of nitrifier communities in marine environments

*Open oceans*

The distribution of nitrifier communities in the open ocean has been widely studied. In this environment, AOA seem to be present throughout the water column and outnumber AOB (Beman et al., 2012; Mincer et al., 2007), which sometimes go undetected (Mincer et al., 2007; Molina et al., 2010). Community distribution changes depending on the area of the ocean. In the Eastern Tropical North Pacific and Gulf of California, AOA are active in the euphotic layer and AOB are confined to higher depths (Beman et al., 2012), while in the North Pacific Subtropical Gyre AOA are predominantly distributed below

the euphotic zone (Mincer et al., 2007). Along the Peruvian coast, both AOA and AOB exhibit a strong *amoA* expression in the upper OMZ (Lam et al., 2009). In the central Pacific Ocean, abundance of the AOA *amoA* gene increases with depth and shows peak abundances in the dimly lit waters of the mesopelagic zone (Church et al., 2010). Moreover, AOA form vertical and latitudinal gradients throughout the North Atlantic, where their abundance and diversity gradually decrease in the meso- and bathypelagic waters from the north towards the equator (Agogué et al., 2008). Shiozaki *et al.* (2016) found that $NH_4^+$

oxidation in the euphotic zone occurred at almost all the stations along a transect from the equatorial Pacific to the Arctic Ocean and was mainly performed by AOA. Studies on nitrifiers in other waters include the Arabian Sea OMZ, where AOA and anammox bacteria have different vertical distributions and occupy different niches (Pitcher et al., 2011a), and the Mediterranean Sea, where the abundance of AOA decreases with depth and is stratified into surface and deep-sea communities (De Corte et al., 2009).

It has also been found that there are two distinct ecotypes of AOA (Francis et al., 2005; Luo et al., 2014). The water column group A (WCA) or "shallow" ecotype is typically most abundant in the epipelagic and upper mesopelagic (Beman et al., 2008), while the water column group B (WCB) or "deep" ecotype dominates the meso- and bathypelagic, where the $NH_4^+$ flux is very low (Sintes et al., 2013). This distribution of AOA ecotypes is consistent with other reports in the Northeast Pacific (Smith et al., 2016) and the California Current (Santoro et al., 2010). Moreover, in the Eastern South Pacific, WCA is

associated with higher $O_2$ and $NH_4^+$ concentrations, while WCB is associated to permanent OMZ and $NH_4^+$-depleted waters (Molina et al., 2010).





Nitrifiers have also been studied in symbiosis with marine organisms. *Nitrospira* seems to be the major driver of nitrification in different types of sponges (Mohamed et al., 2010; Radax et al., 2012)*,* while Zhang et al. (2014a) found that *Nitrosopumilus*-related AOA were the active ammonia oxidizers in four species of sponges. In addition, *Thaumarchaeota* plays an important nitrification role within the tissue of colonial ascidians (Martínez-García et al., 2008).

*Estuaries and coastal environments*

In general, the AOA community is more diverse than that of AOB in estuary sediments (Beman and Francis, 2006; Caffrey et al., 2007; Jin et al., 2011; Zhang et al., 2014b; Zheng et al., 2014). However, the dominance of one group over the other is unclear. In some estuaries, AOA form the most abundant nitrifier community (e.g., Caffrey et al., 2007; Jin et al., 2011; Zhang et al., 2014b), while in others they are less abundant than or equal to AOB (e.g., Mosier and Francis, 2008; Li et al., 2015; Reyes et al., 2017; Zheng et al., 2014). The salinity gradient seems to be the main factor controlling the diversity and distribution of nitrifier communities in estuaries. AOA belonging to "sediment" and "marine" clades (*Nitrosopumilus*-like sequences) are more abundant in the mouth, whereas AOA belonging to the "low salinity" (*Nitrosoarchaeum*-like sequences) and "soil" (group 1.1b) clades are more abundant in the head of many estuaries (Beman and Francis, 2006; Bernhard et al., 2010; Dang et al., 2008; Francis et al., 2003; Mosier and Francis, 2008; Smith et al., 2015b). Similarly, AOB sequences belonging to the *Nitrosospira*-like cluster have mostly been obtained at high salinities, while the *Nitrosomonas*-like cluster is dominant at low salinities (Bernhard et al., 2005; Francis et al., 2003), although both clusters have been found in different parts of the San Francisco Bay estuary (Smith et al., 2015b). In addition, AOA and AOB occupy different niches in the sediments of the estuary; AOB is more abundant where salinity is higher and the C:N ratio is lower, while AOA dominates in the part exhibiting low salinity and a high-C:N ratio (Mosier and Francis, 2008).

Depth, temperature, and $O_2$ play significant roles in determining the community structure of AOA in coastal waters. For instance, WCA organisms are distributed at all depths, whereas WCB organisms are confined to colder, high-nutrient, low-$O_2$ and low-chlorophyll deeper waters along the Chilean and Californian coasts (Bertagnolli and Ulloa, 2017; Smith et al., 2014). In addition, AOA populations fluctuate seasonally, with abundance peaks during winter in the coastal Arctic and North Sea (Christman et al., 2011; Pitcher et al., 2011b). In the Chilean coast, WCB are abundant during spring and summer and are non-detectable during winter (Bertagnolli and Ulloa, 2017).

Ammonia-oxidizing communities have also been studied in coastal microbial mats from the North Sea. In such mats AOB *amoA* genes are significantly more abundant than AOA *amoA* genes, and the composition and abundance of *amoA* genes seem to be driven by salinity, temperature and nutrient concentrations (Fan et al., 2015).

*Deep-sea environments*

Deep-sea sediments and hydrothermal vent systems are sites of active nitrification that harbor diverse and novel ammonia-oxidizing prokaryotes (Dang et al., 2009a; Luo et al., 2015; Nakagawa et al., 2007). In these systems, AOA *amoA* genes are much more diverse but less abundant than the AOB *amoA* genes (Cao et al., 2012; Luo et al., 2015; Nakagawa et al., 2007;





Xu et al., 2014), although several exceptions have been found (Dang et al., 2009a; Yu et al., 2018). Total C and N are suggested as major factors affecting the distribution of AOA and AOB there (Luo et al., 2015; Xu et al., 2014).

*Coral reefs*

The few studies of the nitrifying ecology in coral reefs suggest that AOA may be the main contributors to N cycling in these systems (Beman et al., 2007), with *Nitrosopumilus maritimus* as an important player that could oxidize $NH_4^+$ during the day when the conditions in coral mucus are oxic (Siboni et al., 2008). However, AOB seem to dominate nitrifier communities in other coral species (Yang et al., 2013). Further research in this direction would help understand the importance of both groups involved in the nitrification of a variety of coral species.

**2.3    Comammox**

The process known as comammox was predicted in 2006 (Costa et al., 2006) and finally discovered in 2015 (Daims et al., 2015; van Kessel et al., 2015). The bacterial species performing complete nitrification have been classified as members of lineage II within the genus *Nitrospira*. Daims et al. (2015) reported the discovery of the first comammox bacteria within pipe biofilm at a 1200 m depth in Russia, naming it *Nitrospira inopinata*. Its genome contains *nxr* genes as well as *amo* and *hao*

homologs. van Kessel (2015) also reported the enrichment of two *Nitrospira* species (*Candidatus N. nitrosa* and *Candidatus N. nitrificans*) with similar characteristics to those reported by Daims et al. (2015).

**2.3.1.    Factors affecting comammox and its distribution in marine environments**

As of yet, no comammox bacteria have been found in marine environments. These microorganisms are just beginning to be studied, so the factors controlling comammox and their importance in marine environments are not well understood. It has

been hypothesized that salinity could stop comammox bacteria from thriving in marine ecosystems (Kuypers, 2015), while they could be able to oxidize $NH_4^+$ to $NO_2^-$ under $NH_4^+$ limited conditions and oxidize $NH_4^+$ partially to $NO_2^-$ under oxygen-limited conditions (Kuypers et al., 2018). Kits et al. (2017) discovered that *N. inopinata* has more affinity for $NH_4^+$ than AOA or AOB, showing that comammox bacteria are well adapted to low $NH_4^+$ concentrations. The study also demonstrated that *N. inopinata* has a low affinity for $NO_2^-$, which may prevent it from growing as a pure nitrite oxidizer in environments

where $NO_2^-$ concentration is low (Kuypers, 2017).

**2.4    Dissimilatory nitrate reduction to ammonia (DNRA)**





There are three purposes for $NO_3^-$ reduction: (1) the generation of metabolic energy using the $NO_3^-$ as an electron acceptor ($NO_3^-$ respiration), (2) the dissipation of excess reducing power for redox balance (dissimilatory reduction, DNRA), and (3) the use of $NO_3^-$ as a source of N for growth (assimilatory reduction, ANRA), which will be addressed in the next section.

DNRA can be achieved in two ways: fermentative (reaction a) and chemoautotrophic (reactions b and c). Fermentative

DNRA consists of the reduction of $NO_3^-$ to $NO_2^-$ to produce energy, and then to $NH_4^+$ to allow reoxidation of NADH (Tiedje, 1988). Chemoautotrophic DNRA consists of the reduction of $NO_3^-$ using sulfide ($S^{-2}$), elemental S, or $Fe^{+2}$ as electron donors (Robertson et al., 2016; Slobodkina et al., 2017).

a) $NO_3^- + 2H^+ + 4H_2 \rightarrow NH_4^+ + 3H_2O$

b) $4S + 3NO_3^- + 7H_2O \rightarrow 4SO_4^{-2} + 3NH_4^+ + 2H^+$

c) $8Fe^{+2} + NO_3^- + 21H_2O \rightarrow NH_4^+ + 8Fe(OH)_3 + 14H^+$

In fermentative DNRA, the initial reduction of $NO_3^-$ to $NO_2^-$ occurs in the same way as in denitrification and is catalyzed by either the periplasmic nitrate reductase complex (NapAB) or the membrane-bound nitrate reductase complex (NarGHI) (Moreno-Vivián et al., 1999). The reduction of $NO_2^-$ to $NH_4^+$ is catalyzed by the cytochrome C nitrite reductase (NrfA). Its information is encoded in the *nrfA* gene (Kuypers et al., 2018), which is frequently used as a molecular marker for the whole

DNRA process (Lam et al., 2009; Takeuchi, 2006) (Table S3 in the Supplement). In chemoautotrophic DNRA, the reduction of $NO_3^-$ to $NO_2^-$ is carried out by an enzymatic complex similar to Nap but encoded in the *napMADGH* operon. The reduction of $NO_2^-$ to $NH_4^+$ is catalyzed by alternative enzymes such as the octaheme tetrathionate reductase (Otr) or the octaheme cytochrome C nitrite reductase (Onr) (Kuypers et al., 2018).

A broad diversity of microorganisms is capable of DNRA, mainly prokaryotic organisms belonging to Proteobacteria,

Firmicutes, Verrucomicrobia, Planctomycetes, Acidobacteria, Chloroflexi, and Chlorobia (Tiedje, 1988; Welsh et al., 2014). Marine eukaryotes capable of DNRA include diatoms, which use DNRA to enter a resting stage for long-term survival in dark anoxic sediments (Kamp et al., 2011, 2013), and fungi such as *Aspergillus terreus* isolated from the Arabian Sea OMZ, where it contributes to N loss by fueling anammox (Stief et al., 2014). Fermentative DNRA microbes are favored by non-sulfidic environments with high C:N ratios, whereas chemolithoautotrophic DNRA microbes prefer environments where $H_2S$

is present in appreciable concentrations. Evidence of chemolithotrophic DNRA in marine sediments has been found in *Beggiatoa*, which reduces $NO_3^-$ using $S^{-2}$ as an electron donor (Preisler et al., 2007), as well as in *Thermosulfurimonas dismutans* and *Dissulfuribacter thermophilus* isolated from deep-sea hydrothermal vents; these are thermophilic anaerobic bacteria that grow autotrophically with elemental S as an electron donor and $NO_3^-$ as an electron acceptor (Slobodkina et al., 2017).

In contrast to denitrification or anammox, DNRA produces $NH_4^+$ without the release of $N_2O$ or $N_2$, thus does not cause N loss and contributes to primary production and nitrification. Therefore, understanding the mechanisms controlling the DNRA community and its interactions with denitrification and anammox communities is critical for understanding the fate of N in marine systems.




### 2.4.1    Factors affecting DNRA in marine systems

DNRA and denitrification compete for $NO_3^-$ and there are several factors favoring DNRA over denitrification, such as high $S^{-2}$ concentrations and $C:NO_3^-$ ratios, elevated temperatures, salinity and anoxic conditions (Song et al., 2014). Although DNRA has a lower energetic yield than denitrification, it can accept a greater number of electrons per $NO_3^-$ molecule (eight,

compared to five for denitrification). For this reason, DNRA may be energetically favored over denitrification in C-rich anoxic environments where $NO_3^-$ is limiting and electron donors (organic C or $S^{-2}$) are in excess (Kraft et al., 2014; Tiedje, 1988). It has been suggested that DNRA is regulated by $O_2$ and is not affected by $NO_2^-$ (van den Berg et al., 2017), which contrasts with reports of DNRA rates being stimulated by $NO_2^-$ in estuaries (Yin et al., 2017). It is also assumed that DNRA is not affected by $NH_4^+$ (Papaspyrou et al., 2014; Tiedje, 1988), but it has been demonstrated that $NH_4^+$ concentration is

positively correlated with DNRA rates (Lisa et al., 2015; Song et al., 2014). Salinity seems to be another factor affecting DNRA, although there is not a clear pattern. In some estuaries DNRA activity grows with increasing salinity (Gardner et al., 2006; Giblin et al., 2010; Lisa et al., 2015), while in others increasing salinity causes DNRA to decrease (Dong et al., 2009). The abundance of *nrfA* genes is correlated with $S^{-2}$ and $Fe^{2+}$, which can enhance DNRA by providing extra free energy (Robertson et al., 2016; Yin et al., 2017). Elevated temperatures increase DNRA rates; therefore, DNRA may be an

important pathway for $NO_3^-$ conversion during summer (Yin et al., 2017). In coastal zones, this process can also be affected by physical processes such as tides, which transport $NO_3^-$ and support the conversion (Zheng et al., 2016).

### 2.4.2    Distribution of DNRA communities in marine environments

Since DNRA is an anaerobic process, marine DNRA communities are mostly restricted to anoxic environments such as sediments and OMZ. However, there are relatively few studies on DNRA communities in those marine systems compared to

the number of studies on microbes involved in other N-cycling processes (Damashek and Francis, 2017; Lam and Kuypers, 2011).

*Open oceans and deep-sea environments*

Only a few studies on DNRA distribution have been conducted in open oceans, most of them in OMZ. DNRA is coupled

with anammox in the Arabian sea OMZ, as indicated by $^{15}N$ experiments (Jensen et al., 2011), and is carried out by microorganisms carrying divergent *nrfA* genes (Lüke et al., 2016). DNRA has also been detected in the Peruvian OMZ, where it supplies most of the $NH_4^+$ needed for anammox (Lam et al., 2009). However, Kalvelage et al. (2013) reported a lack of detectable *nrfA* genes and low DNRA rates in that zone and concluded that the process is sporadic. Moreover, DNRA communities have also been found in deep-sea sediments such as those from the South China Sea, where microbes involved

in DNRA, ammonia oxidation and anammox are dominant (Yu et al., 2018). More studies are needed to confirm the importance of the DNRA process in deep-sea environments.



*Estuaries and coastal zones*

Diversity of the *nrfA* gene in estuaries is high and often changes along salinity and $NO_3^-$ gradients (Smith et al., 2007; Song et al., 2014; Takeuchi, 2006). There are exceptions such as in the Yellow River estuary, where the structure of DNRA communities is not affected by either type of gradient (Bu et al., 2017). In addition, DNRA communities from estuarine
sediments are site-specific and could vary significantly at a small spatial scale (Decleyre et al., 2015). For example, the abundance of the *nrfA* gene varies vertically in the sediments of the Yangtze estuary (China) and is greater than anammox or denitrifying genes in deeper sediments (Zheng et al., 2016). In the well-studied Colne estuary (UK), DNRA communities embedded in deeper anaerobic sediments are more homogeneous compared to those in the surface (Takeuchi, 2006). Furthermore, DNRA rates and abundance of the *nrfA* gene are higher at the head of that estuary, where the $NO_3^-$
concentration is higher (Dong et al., 2009; Smith et al., 2007). Finally, higher abundance of the *nrfA* gene and DNRA activity have been found in estuarine sediments richer in organic C and $S^{-2}$ (Song et al., 2014).

## 2.5    Assimilatory nitrate reduction to ammonia (ANRA)

The first step of ANRA is the reduction of $NO_3^-$ to $NO_2^-$, and then to $NH_4^+$. This process is summarized in the following reactions:

$NAD(P)H + H^+ + NO_3^- + 2e^- \rightarrow NO_2^- + NAD(P)^+ + H_2O$

$6Ferredoxin\ (red) + 8H^+ + 6e^- + NO_2^- \rightarrow NH_4^+ + 6\ Ferredoxin\ (oxi) + 2H_2O$

ANRA is performed by two classes of cytoplasmic assimilatory nitrate reductases (Nas): ferredoxin-dependent Nas (or flavodoxin-dependent Nas) and NADH-dependent Nas, which are encoded in the *nas* genes (Moreno-Vivián et al., 1999). $NO_3^-$ is the major N source for marine eukaryotes, bacteria and archaea that contain *nas* genes (Moreno-Vivián et al., 1999).

### 2.5.1    Factors affecting ANRA and its distribution in marine environments

The few studies on the diversity of the *nasA* gene have revealed that nitrate-assimilating bacteria (NAB) are widely distributed in oceanic environments (Table S4 in the Supplement). In general, temperature, $NO_3^-$ concentration and salinity are the main factors shaping the marine NAB communities (Allen et al., 2005; Jiang et al., 2015; Jiang and Jiao, 2016). For example, marine NAB community structure and abundance are positively correlated with $NO_3^-$ concentration (Allen et al.,
2005). In addition, it has been demonstrated that $NH_4^+$ acts as a repressor of *nas* genes in isolated strains (Lledó et al., 2005). Proteobacteria, Bacteroidetes and Cyanobacteria are the dominant *nasA*-harboring bacteria in the South China Sea, Indian Ocean, and Pacific Ocean (Cai and Jiao, 2008; Jiang et al., 2015). The composition and diversity of NAB communities vary from one location to another and seem to be affected by salinity. For example, NAB communities in the South China Sea are more diverse than those in the Indian Ocean, while coastal NAB communities are less diverse than those from open waters.
In addition, chemolithoautotrophic sulfur-oxidizing bacteria are the primary NAB in the Kueishantao hydrothermal vent,





indicating a unique NAB community in that extreme environment (Jiang et al., 2015). Another study in the Indian Ocean indicates that NAB communities are dominated by two Gammaproteobacteria with different vertical distribution: *Vibrio* dominates in the euphotic layer, while *Marinobacter* does so at higher depths (Jiang and Jiao, 2016). Other NAB communities have been reported in the South Atlantic Bight, the North Pacific Gyre, Monterey Bay, Tampa Bay, the

Norwegian coast, and the Barents Sea (Adhitya et al., 2007; Allen et al., 2001, 2005; Jenkins et al., 2006).

### 2.6    Denitrification

Denitrification is a canonical, facultative, anaerobic respiratory pathway performed by a diverse array of microorganisms in which $NO_3^-$ is respired to $NO_2^-$, followed by stepwise reductions to NO, $N_2O$, and $N_2$ (Zumft, 1997). Denitrification is distributed within a taxonomically diverse group of microorganisms that can have different combinations of genes involved

in the denitrification pathway (Jones et al., 2008).

The reduction of $NO_3^-$ to $NO_2^-$ is catalyzed by two enzymes: NarGHI, present in a wide variety of microorganisms, and NapAB, restricted to Gram-negative bacteria (Philippot et al., 2002). The active sites of both complexes are encoded in *narG* and *napA* genes, which have been frequently used as biomarkers for the microorganisms involved in this process in marine systems (Lam et al., 2009; Smith et al., 2007) (Table S5 in the Supplement). Nitrate reduction is a major source of $NO_2^-$ for

other N-cycling processes, including aerobic nitrite oxidation and anammox (Kuypers et al., 2018).

Two isofuctional periplasmic enzymes catalyze the reduction of $NO_2^-$ to NO: a Cu-containing nitrite reductase (encoded by *nirK*) and a haem-containing cd1 nitrite reductase (encoded by *nirS*), with most denitrifying bacteria containing one of the two enzymes (Zumft, 1997). Because this is the first committed step of the pathway to a N gaseous product, *nir* genes are the most widely used markers for denitrifiers (Braker et al., 2000; Mosier and Francis, 2010; Ward et al., 2009) (Table S5 in the

Supplement). Notwithstanding, these genes are present in many other microorganisms, including anammox bacteria, nitrite and methane-oxidizing bacteria, AOA and AOB (Kuypers et al., 2018). The taxonomic diversity of *nirK*-denitrifiers is greater than that of *nirS*-denitrifiers, likely due to horizontal gene transfer. Furthermore, *nirS*-denitrifiers seem to have a complete denitrification pathway (including *nor* and *nos* genes); thus, they are more likely to completely reduce $NO_2^-$ to $N_2$ (Graf et al., 2014). In addition, the distribution of *nir*-containing denitrifiers is environment-specific and seems to be

controlled by redox conditions, with *nirS* communities prevailing under lower dissolved oxygen regimes (Kim et al., 2011; Pajares et al., 2017).

The conversion of NO to $N_2O$ is carried out by two types of nitric oxide reductases (NOR): a cytochrome c-dependent complex (cNOR) present almost exclusively in denitrifiers, and a quinol-dependent complex (qNOR) present in microorganisms with incomplete denitrifying molecular machinery (Hino et al., 2010). cNOR is the most studied NOR and

is composed of the subunits NorB and NorC. NorB is encoded in the homonymous gene, which is present in denitrifying bacteria and several AOB (Casciotti and Ward, 2005); it has been used as a biomarker for the microorganisms involved in the reduction of NO (Braker and Tiedje, 2003).





The final step in the denitrification pathway is the reduction of $N_2O$ to $N_2$. It is catalyzed by the nitrous oxide reductase (NOS), which is encoded in the *nosZ* gene and is frequently used as a biomarker for the study of denitrifier communities capable of reducing $N_2O$ in marine systems (Bowles et al., 2012; Castro-González et al., 2015; Wittorf et al., 2016) (Table S5 in the Supplement). There are two phylogenetically distinct *nosZ* clades: clade I includes organisms with a complete

denitrification pathway, whereas clade II includes organisms that frequently lack other denitrification genes (Jones et al., 2013).

### 2.6.1 Factors affecting denitrification in marine systems

Oxygen is the main driver of denitrifier communities in marine systems, since denitrification is limited to environments where $O_2$ is nearly fully depleted (<5 μmol $O_2$/L). It has been reported that *nirS, norB* and *nosZ* genes strongly decrease in

$O_2$ concentrations >200 nM (Dalsgaard et al., 2014). For instance, the *nirS* gene is rarely present in well-oxygenated waters or waters containing $H_2S$, and is found at high concentrations in the Arabian Sea OMZ (Jayakumar et al., 2004; Ward et al., 2009). Furthermore, $O_2$ availability is associated with the habitat partitioning of $N_2O$ reducers in coastal marine surface sediments (Wittorf et al., 2016). Other factors affecting marine denitrification vary from one environment to another. For example, in some OMZs, such as those in the Eastern Tropical North and South Pacific, denitrification appears to be limited

by organic C, while in others, such as the Arabian Sea OMZ, is unaffected by it (Ward et al., 2008). In the OMZs of the Eastern South Pacific and the Mexican Pacific coast, denitrifying communities are affected by $NO_3^-$ and $NO_2^-$ (Castro-González et al., 2005; Liu et al., 2003). Organic matter and salinity are among the key factors controlling denitrification rates, as well as denitrifier community abundance and composition in estuaries (Francis et al., 2013, Lee and Francis 2017; Mosier and Francis, 2010; Zhang et al., 2014b). The clay percentage and pH are other important factors influencing the

abundance of *nir* genes in intertidal sediments (Wang et al., 2014). Since NosZ requires copper, it could represent a regulating factor in denitrification and the production of $N_2O$ in marine environments (Granger and Ward, 2003). As mentioned before, denitrification and DNRA compete for $NO_3^-$, and factors such as low $S^{-2}$ concentrations (which inhibit the last two steps of denitrification), cold temperatures and a low $C:NO_3^-$ ratio favor denitrification over DNRA (Burgin and Hamilton, 2007; Song et al., 2014).

### 2.6.2 Distribution of denitrifier communities in marine environments

Although denitrifying microorganisms can be found in any marine environment, denitrification is typically restricted to suboxic or anoxic environments such as OMZs and sediments.

*Open oceans and deep-sea environments*





OMZs are considered one of the major oceanic sites of denitrification. For example, denitrification is the dominant N-loss pathway in the Arabian Sea OMZ (Ward et al., 2009), where denitrifiers dominate over anammox bacteria (Jayakumar et al., 2004; Jayakumar et al., 2009). Depth distributions of *nir* genes follow the same pattern in the Arabian Sea and the Eastern Tropical South Pacific, where they are associated with the secondary $NO_2^-$ maximum in oxygen-depleted waters. However,

the denitrifier community composition of the two sites seems to be different (Castro-González et al., 2005; Jayakumar and Ward, 2013). In the Eastern Tropical North Pacific, *narG* seems to be the most abundant denitrifying gene, while *nirK* dominates over *nirS* and the two *nosZ* clusters based on depth distributions corresponding to two $N_2O$ peaks (Fuchsman et al., 2017). Unlike *nirS*-based communities in the Black Sea suboxic zone, *nirK*-based communities generally vary with depth, while the composition of both *nirK* and *nirS* genes changes dramatically at the bottom of this suboxic zone (Oakley et

al., 2007). Great genomic potential for full denitrification to $N_2$ has been found in Bothnian Sea sediments located in the northern Baltic Sea, with higher abundance of *narG*-, *nirS*- and *nosZ*-encoding gene reads but minor importance of genomic potential for anammox and DNRA (Rasigraf et al., 2017). In addition, the composition of *nirS*-based communities is site-specific in the Baltic Sea and varies along biogeochemical gradients ($H_2S$, $NH_4^+$, $NO_3^-$ and $O_2$) in the water column, while it is uniform in the sediment (Falk et al., 2007; Hanning et al., 2006).

Few studies of denitrifier communities have been conducted in deep-sea sediments or hydrothermal vents, where chemolithotrophic denitrification is expected to be an important process given the high concentration of reduced sulfur species in such environments. For example, *nirS* sequences have been retrieved from a hydrothermal vent system in the Juan de Fuca Ridge (Bourbonnais et al., 2014) and from deep-sea sediments in the Nankai Trough, where *nirK*-type denitrifiers were undetected (Tamegai et al., 2007).

*Estuaries and coastal environments*

Denitrification is often the major process driving N removal from coastal and estuarine environments (Damashek and Francis, 2017; Devol, 2015). Sediments provide ideal conditions for this process, due to the narrow spatial scale for diffusion across redox boundaries. Therefore most studies have documented the abundance or diversity of *nir* genes in estuary

sediments, with a few of them in estuary waters (Zhang et al., 2014b). The dominant denitrifying gene ecotypes identified in these systems are not affiliated with known denitrifying strains (Francis et al., 2013; Lee and Francis, 2017; Santoro et al., 2006). In general, *nirS* genes are more abundant and diverse than *nirK* in estuary sediments (e.g., Abell et al., 2010; Lee and Francis, 2017; Magalhães et al., 2011; Mosier and Francis, 2010; Santoro et al., 2006; Smith et al., 2015a; Wang et al., 2014). For example, *nirS*-type denitrifiers dominate in shallow bay sediments from the South China Sea, whereas *nirK*-type

denitrifiers go undetected (Yu et al., 2018). Diversity and abundance of denitrifiers often change along the estuarine salinity gradient, with distinct communities in fresh and marine regions (Abell et al., 2013; Francis et al., 2013; Lee and Francis, 2017). For instance, in the San Francisco Bay estuary, the abundance of *nirK* genes seems to be higher in the riverine zone whereas *nirS* genes are more abundant in zones that are more marine (Mosier and Francis, 2010). Additionally, nitrate and




nitrite reductase gene copy numbers decline significantly from the head towards the mouth of the Colne estuary (Smith et al., 2007).

It is worth noting that denitrification has also been proved to happen in particular marine environments and organisms. For example, it has been shown that *Trichodesmium* harbors denitrifier communities composed by Alphaproteobacteria that

actively express *nosZ*, showing that *Trichodesmium* colonies are potential sites of $N_2O$ reduction (Coates and Wyman, 2017). In addition, sinking copepod carcasses have anoxic interiors that support the expression of *nirS* genes, representing hotspots of pelagic denitrification (Glud et al., 2015). Finally, certain benthic foraminifera appear to be capable of accumulating and respiring $NO_3^-$ through denitrification (Piña-Ochoa et al., 2010; Risgaard-Petersen et al., 2006).

**2.7 Nitrate/Nitrite-dependent anaerobic methane oxidation**

Anaerobic methane oxidation coupled with denitrification, also known as nitrate/nitrite-dependent anaerobic methane oxidation (n-damo), was discovered in 2006 and constitutes a unique link between the two major global nutrient cycles of C and N (Raghoebarsing et al., 2006). The process is given by the following reactions:

(1) $CH_4 + 4NO_3^- + 8H^+ \rightarrow CO_2 + 4NO_2^- + 10H_2O$

(2) $3CH_4 + 8NO_2^- + 8H^+ \rightarrow 3CO_2 + 4N_2 + 10H_2O$

The n-damo process is carried out by bacteria from the NC10 phylum and ANME archaea in a syntrophic manner. *Candidatus Methanoperedens nitroreducens* (ANME-2d) is one of the microorganisms capable of performing reaction (1) (Haroon et al., 2013), while *Candidatus Methylomirabilis oxyfera* (a member of the NC10 phylum) is capable of performing reaction (2) (Ettwig et al., 2010). The metabolism of *M. nitroreducens* is highly complex, and includes genes related to

reverse methanogenesis (Haroon et al., 2013; Timmers et al., 2017). The metabolism of *M. oxyfera* is also complex and unusual, because in spite of being considered an anaerobic microorganism, it is able to oxidize methane ($CH_4$) using enzymes found in aerobic methanotrophs (Ettwig et al., 2010) such as the particulate methane-monooxygenase (pMMO), encoded in the *pmoA* gene, which has been used as a biomarker in marine systems (Chen et al., 2016; Padilla et al., 2016) (Table S6 in the Supplement). The metabolism of *M. oxyfera* has been described as "intra-aerobic" and consists of the

intracellular production of $O_2$ by dismutating NO into $O_2$ and $N_2$, catalyzed by the nitric oxide dismutase (NOD) (Ettwig et al., 2010). The genome of *M. oxyfera* also encodes NirS and three qNOR, which may participate in detoxifying processes (Wu et al., 2011).

**2.7.1 Factors affecting n-damo and its distribution in marine systems**

The n-damo process has just begun to be studied, and only a few works on the distribution of these microorganisms in

marine environments and the factors affecting their distribution are available. In oceanic environments, n-damo has been detected in OMZs, which represent a niche for NC10 bacteria (Chronopoulou et al., 2017; Padilla et al., 2016). For example,





in the OMZ off northern Mexico and Costa Rica, 16S rRNA showed a peak abundance of NC10 bacteria in the anoxic zone with high $NO_3^-$ and $CH_4$ concentrations (Padilla et al., 2016).

*M. oxyfera*-like bacteria have also been detected in estuaries. In the sediments of Zhangjiang estuary, *M. oxyfera*-like bacteria show great diversity and a depth-specific distribution influenced by redox potential, water content and total organic

C (Zhang et al., 2018). In the Jiaojiang estuary, organic content seems to be the most important factor affecting the distribution of n-damo communities, NC10 bacteria being more abundant in the intertidal than in the subtidal sediments (Li-dong et al., 2014). In addition, the activity of NC10 bacteria varies seasonally and spatially in the coastal sediments of the East China Sea, being higher in the spring and lower in the intertidal zone (Wang et al., 2017). The structure of this community in coastal sediments seems to be highly influenced by $NO_3^-$ concentrations (Wang et al., 2017), as well as by

$NH_4^+$ and $NO_2^-$ (Chen et al., 2015).

In general, n-damo microorganisms are favored by low $SO_4^{-2}$ concentrations (due to the elimination of competition with $SO_4^{-2}$ reducers) and high $CH_4$ concentrations. In addition, it is believed that the abundance of these microorganisms is controlled by $NO_2^-$ due to possible competition with denitrifying and anammox microorganisms. If anammox bacteria are not limited by $NH_4^+$ concentrations, they are more competent for $NO_2^-$ than n-damo microorganisms (Luesken et al., 2011). The

importance of n-damo microorganisms in the C and N cycles, in addition to the small number of studies on their distribution, clearly warrant further study to ascertain the drivers of these communities in different marine ecosystems.

### 2.8   Anammox

The anaerobic oxidation of ammonium, anammox, consists in the conversion of $NH_4^+$ and $NO_2^-$ to $N_2$ (4) in the absence of $O_2$, and involves three reactions (1-3):

(1)   $NO_2^- + 2H^+ + e^- \rightarrow NO + H_2O$

(2)   $NH_4^+ + NO + 2H^+ 3e^- \rightarrow N_2H_4 + H_2O$

(3)   $N_2H_4 \rightarrow N_2 + 4H^+ + 4e^-$

$$\overline{\phantom{xxxxxxxxxxxxxxxxxxxxxxxxx}}$$

(4)   $NH_4^+ + NO_2^- \rightarrow N_2 + 2H_2O$

Reaction (1) is catalyzed by the NirS, while reaction (2) is carried out by the hydrazine synthase (HZS). Enzyme-catalyzing reaction (3) has been called many names over the years, such as hydrazine oxidase (HZO) or hydrazine dehydrogenase (HDH) (Kartal et al., 2011). Various functional genes have been used as anammox biomarkers in marine systems (Table S7 in the Supplement): *Scalindua*-like *nirS*, which codes for a NirS specific to *Scalindua*, the dominant anammox bacteria in marine OMZs (Lam et al., 2009; Li et al., 2013); *hzoAB* (Hirsch et al., 2011; Lisa et al., 2014), which codes for part of the

HZO, and with numerous divergent copies in a number of anammox bacteria (Strous et al., 2006); and *hzsA* or *hzsB*, coding for part of the HZS, which have been suggested as the most suitable biomarkers for the process (Han et al., 2017; Harhangi et al., 2012; Nunoura et al., 2013).



Until now, ten *Candidatus* species belonging to five genera have been reported as responsible for anammox, all of them within a deep, monophyletic branch in the order Planctomycetales: *Kuenenia*, *Anammoxglobus* and *Jettenia* with one species each, *Brocadia* with three species, and *Scalindua* with four (Kartal et al., 2012; Van de Vossenberg et al., 2013). Anammox bacteria are autotrophs, active at a wide range of temperatures (1.8°C – 85°C) and with great affinity for $NO_2^-$

and $NH_4^+$, even in concentrations below 5 μM (Jetten et al., 2009). These slow-growing anaerobic microorganisms are difficult to culture and possess a unique capability of producing and converting $N_2H_4$ in a ladderane lipid membrane called the anammoxosome (Kartal et al., 2012).

### 2.8.1    Factors affecting anammox in marine systems

Although the presence of anammox bacteria may not be indicative of high anammox activity, several studies have shown

that anammox bacterial gene abundance positively correlates with anammox rates in marine environments (Bale et al., 2014; Hou et al., 2013; Lisa et al., 2014). In general, anammox activity is mainly regulated by $O_2$ and inorganic N concentrations and seems to be coupled with nitrification (Lam et al., 2007), denitrification (Dalsgaard et al., 2003) and DNRA (Jensen et al., 2011). Despite being described as obligated anaerobes active only at $O_2$ concentrations below 2 μM, anammox bacteria are resistant to $O_2$ exposure; nevertheless, the process is inhibited at high $O_2$ concentrations (Jetten et al., 2009). It has been

assumed that anammox bacteria in OMZs may be more frequently limited by $NH_4^+$ than by $NO_2^-$, because $NH_4^+$ concentrations in OMZs are typically lower than those of $NO_2^-$ (Lam et al., 2009). However, studies have shown that $O_2$ and $NO_2^-$ co-limit the distribution of anammox bacteria in the OMZ of the Black Sea (Lam et al., 2007), Arabian Sea (Pitcher et al., 2011) and Eastern Tropical North Pacific (Kong et al., 2013; Rush et al., 2012). It has been also reported that the availability of $NO_3^-$, $NO_2^-$, and $NH_4^+$ regulates anammox activity in coastal and estuary sediments (Nicholls and Trimmer,

2009; Risgaard-Petersen et al., 2004; Teixeira et al., 2016; Trimmer et al., 2005), in which the fluctuating availability of $O_2$, $NO_3^-$ and $NO_2^-$ typically favors denitrifying microorganisms over anammox bacteria (Risgaard-Petersen et al., 2005). In addition, in many marine environments anammox activity and bacterial distribution are highly dependent upon organic C content (Shehzad et al., 2016; Trimmer et al., 2003), salinity (Hou et al., 2013; Rich et al., 2008; Shehzad et al., 2016; Sonthiphand et al., 2014) and temperature (Rysgaard-Petersen et al., 2004; Shehzad et al., 2016). For example, in eutrophic

environments, high organic C concentrations usually stimulate denitrification while suppressing anammox because of the competition for $NO_2^-$ (Brin et al., 2014; Jäntti et al., 2011; Nicholls and Trimmer, 2009). However, a number of studies have found positive correlations between organic C content in marine sediments and anammox rates caused by high production of $NH_4^+$ or $NO_2^-$ from heterotrophic remineralization and nitrification (Bale et al., 2014; Lisa et al., 2015; Trimmer et al., 2003). Furthermore, anammox activity has been found to be strongly correlated with the sinking of organic matter in the

Eastern Tropical South Pacific OMZ (Kalvelage et al., 2013). These contradictions show that understanding the relationship between organic C content and anammox is still an open question. Finally, anthropogenic contaminants in marine





environments, such as heavy metals, have also been identified as key factors influencing the ecology and biogeochemical functioning of anammox bacteria (Dang et al., 2013).

### 2.6.2 Distribution of anammox bacteria in marine environments

Anammox bacteria contribute to approximately 30–60% of marine N loss and are present and active in a wide range of
marine environments, especially in oxygen-depleted settings such as the water column and OMZ sediments (e.g., Galán et al., 2009; Kuypers et al., 2003; Jensen et al., 2011; Lam et al., 2007; Ward et al., 2009), eutrophic bays (e.g., Dang et al., 2010; Lisa et al., 2014), estuarine sediments (e.g., Brin et al., 2014; Li et al., 2011; Tal et al., 2005; Trimmer et al., 2005), fjord sediments (Brandsma et al., 2011; Risgaard-Petersen et al., 2004), Arctic sediments (Rysgaard et al., 2004), deep ocean sediments (Hong et al., 2011a, 2011b; Shao et al., 2014), and deep-sea hydrothermal vents (Byrne et al., 2009). Below, we
summarize the main findings from a large number of studies on the distribution of anammox bacteria in the most representative marine environments.

*Open oceans*

The release of $NH_4^+$ and $NO_2^-$ through incomplete denitrification and deficient dissolved oxygen makes OMZs ideal
environments for the growth of anammox bacteria. These microorganisms are found in different OMZs such as those in the Baltic Sea (Kuypers et al., 2003), Golfo Dulce (Dalsgaard et al., 2003), Eastern Tropical North Pacific (Kong et al., 2013; Rush et al., 2012), Eastern Tropical South Pacific (Galan et al., 2009), Colombian Pacific (Castro-González et al., 2014), Arabian Sea (Jaeschke et al., 2007; Jensen et al., 2011; Ward et al., 2009) and Benguela (Kuypers et al., 2005), where the anammox process accounts for between one-fifth and all of $N_2$ production (Dalsgaard et al., 2005). In fact, anammox is the
dominant N-loss pathway in the Benguela upwelling system (Kuypers et al., 2005), the Black Sea (Lam et al., 2007) and the Peruvian OMZ (Hamersley et al., 2007). A low diversity of anammox communities has been detected in OMZ waters (Kong et al., 2013; Schmid et al., 2007), in which mainly two clades of *Ca. Scalindua* typically predominate: Clade 1 (*Ca. Scalindua sorokinii/brodae*) in the Peruvian and Namibian OMZs, and clade 2 (*Ca. Scalindua arabica*) in the Peruvian and the Arabian Sea OMZs (Woebken et al., 2008). Another study in the Baltic Sea suboxic zone indicates that *Ca. Scalindua*
species split into two clusters in that environment: *Ca. Scalindua richardsii*, present in the upper suboxic zone at high $NO_2^-$ and $NO_3^-$ and low $NH_4^+$ concentrations, and *Ca. Scalindua sorokinii*, present in the lower suboxic zone at high $NH_4^+$ and low $NO_3^-$ concentrations (Fuchsman et al., 2012). Sediments from the north marginal seas in China harbor two potential novel anammox bacteria, with *Ca. Scalindua* being the most abundant in said environments (Shehzad et al., 2016).

*Estuaries and coastal environments*

Anammox has been reported in different estuarine environments, such as the Thames estuary (Trimmer et al., 2003), the New England estuary (Brin et al., 2014), the New River estuary (Lisa et al., 2015), the Randers Fjord (Risgaard-Petersen et



al., 2004), and the Yangtze estuary (Hou et al., 2013), where anammox bacterial diversity and distribution are mainly affected by salinity gradients (Sonthiphand et al., 2014). Other reports of the presence of anammox bacteria comprise coastal areas such as the Inner Harbor (Tal et al., 2005), and coastal sediments from Europe, North America (Engström et al., 2005) and the Bohai Sea (Dang et al., 2013). *Ca. Scalindua* typically dominates throughout estuarine sediments (Dang et al., 2010;

Hirsch et al., 2011; Rich et al., 2008; Tal et al., 2005), while *Ca. Brocadia, Ca. Kuenenia, Ca. Anammoxglobous* and *Ca. Jettenia* are mainly found in fresh to oligohaline sediments (Dale et al., 2009; Hirsch et al., 2011).

*Deep-sea and other environments*

Deep-sea and other extreme environments seem to harbor a great diversity of anammox species. For example, a high
diversity of *Ca. Scalindua*, grouped into five clusters, was found in deep-sea sediments from the South China Sea (Hong et al., 2011a). Anammox bacteria are also present and active in hydrothermal vent areas, such as those in the Mid-Atlantic Ridge, where *Ca. Kuenenia* was found at 300°C (Byrne et al., 2009), and the Okhotsk Sea, where *hzo* genes are highly abundant (Shao et al., 2014). A study in the Guaymas Basin revealed that *Ca. Scalindua* species are more abundant in cold hydrocarbon-rich sediments than hydrothermal vents (Russ et al., 2013).

**3    Effects of anthropogenic activity on the marine nitrogen cycle**

The marine N cycle is currently being largely perturbed by human activity. Recent findings show that the oceanic N budget balance leans more significantly towards higher losses than inputs (Codispoti et al., 2001; Gruber and Galloway, 2008; Voss et al., 2013). Anthropogenic activities pertaining to the production of artificial fertilizers and fossil fuel combustion are mainly responsible for this imbalance, affecting the marine N cycle directly or indirectly (Fig. 3). Direct alterations include
the input of fixed N through riverine discharges and atmospheric deposition (Duce et al., 2008; Jickell et al., 2017). Large anthropogenic N inputs cause eutrophication and the formation of anoxic or hypoxic areas ("dead zones"), impacting primary production and the marine trophic web (Vaquer-Sunyer and Duarte, 2008) with consequent ecosystemic collapse (Diaz and Rosenberg, 2008). Indirect alterations include activities that increase the atmospheric concentration of greenhouse gases, leading to ocean warming, acidification and deoxygenation (Hutchins and Fu, 2017). The scale and impact of such
anthropogenic perturbation of the marine N cycle remain highly uncertain (Gruber, 2016; Hutchins and Fu, 2017).

Estimated anthropogenic release of N into the global environment ($\sim$160 Tg N yr$^{-1}$) is now of similar magnitude to all natural global ocean $N_2$ fixation ($\sim$140 Tg N yr$^{-1}$), and may increase along with growing global population (Eugster and Gruber, 2012; Gruber and Galloway, 2008). Modeling studies have shown that the predicted anthropogenic increase in atmospheric N deposition could initiate a series of strong negative feedback in the marine N cycle. There could be a pronounced decrease
in global ocean $N_2$ fixation and an increase in denitrification, which would compensate for much of the enhanced N input and limit the impact of atmospheric N deposition on marine productivity (Somes et al., 2016; Yang and Gruber, 2016).



Ocean warming enhances water stratification and decreases oxygen solubility, causing a major loss of $O_2$ from the future ocean (Dybas, 2005). Worldwide expansion of dead zones and OMZs will have large impacts on microbially mediated N processes in marine ecosystems. The hypoxic conditions of these environments may favor $N_2$ fixation (since nitrogenase is strongly inhibited by $O_2$; Sohm et al., 2011) and anaerobic N cycling processes, such as denitrification and anammox, since

N isotopic labeling analyses suggest that both processes are strongly negatively correlated with $O_2$ concentration (Kalvelage et al., 2011; Neubacher et al., 2013; Ward et al., 2009). Global proliferation of suboxic waters could also promote nitrification, since this process occurs at high rates in transitional regions around OMZs where $O_2$ is low but not fully depleted (Lam et al., 2009). Furthermore, ocean warming will likely cause an increase of $N_2$ fixation, since enzyme activity increases at higher temperatures and higher ocean surface temperatures will lead to an expansion of habitats suitable for

diazotrophs (Hutchins et al., 2009). An increase in marine $N_2$ fixation would therefore lead to an increase in the amount of Nr available for further processes in the N cycle.

Ocean acidification may stimulate $N_2$-fixation rates and reduce nitrification rates within coming decades. A number of studies have documented the positive effects of elevated $CO_2$ on marine $N_2$-fixing cyanobacteria (Hutchins et al., 2007; Lomas et al., 2012; Rees et al., 2017) but others did not find a clear response (Böttjer et al., 2014; Law et al., 2012), possibly

due to regional and taxonomic differences or other associated environmental factors. For instance, a decrease in pH due to ocean acidification may lead to a decrease in the bioavailability of Fe (Shi et al., 2010), which may in turn lead to a decrease in $N_2$ fixation in areas where Fe is limiting. On the other hand, acidification experiments have demonstrated decreases in $NH_3$ oxidation due to the incremental protonation of $NH_3$ (the substrate for microbial ammonia oxidizers) to $NH_4^+$ as seawater pH decreases (Beman et al., 2011; Kitidis et al., 2011).

Denitrification and nitrification processes are responsible for 87% of the annual global $N_2O$ budget (18.8 Tg N yr$^{-1}$; Syakila and Kroeze, 2011), with a 30% contribution from the world's oceans (Voss et al., 2013). The heat-trapping capacity of $N_2O$ is 300 times that of $CO_2$ (Rodhe, 1990), thus it plays a key role in the greenhouse effect and climate regulation. Additionally, $N_2O$ acts as a source of NO and N dioxide ($NO_2$), gases involved in ozone ($O_3$) depletion in the stratosphere and acid rain formation (Ravishankara et al., 2009). $O_3$ depletion results in increased UVB irradiance that threatens nutrient cycling,

primary production, community diversity and species composition by harming phytoplankton and other organisms (El-Sayed et al., 1996 and references therein). Consequently, the marine pathways of $N_2O$ and the quantification of its oceanic emissions have recently received increased attention (Babbin et al., 2015; Freing et al., 2012).

Finally, N cycling is connected to the cycles of biologically important elements such as those of C and phosphorous, which implies that human alterations of marine N cycling are likely to have major consequences for other biogeochemical

processes and affect marine ecosystem functions and services. However, the understanding of these consequences is poor due to the lack of globally empirical studies and sufficiently global models (Fowler et al., 2015).

### 4    Conclusions and perspectives



In this review, we have provided a panorama of the biochemistry, genetics, ecology and distribution of marine N-cycling microbes and the processes they mediate. These microorganisms are more widely distributed than previously thought, given that they have been found in unpredicted marine environments. Plus, many new metabolic pathways for N compounds have been reported over the past few years, completely changing the paradigm of the classic marine N cycle. The activity and

distribution patterns of N-cycling microbes in marine ecosystems seem to be dictated by site-specific factors, of which temperature, salinity, $O_2$ and inorganic N forms are among the main factors controlling these communities in most cases. Additionally, we have examined the potential effects of human activity on microbially mediated N-cycling processes; such activity has led to an alteration of the natural balance of the marine N cycle, with consequences that we are just beginning to experience and comprehend.

We have also identified large knowledge gaps regarding microbially mediated N processes in marine ecosystems, and these represent potential areas of exploration. Future research should focus on: 1) detecting novel N-cycling processes, such as comammox and n-damo, in marine ecosystems and taking steps to further understand the physiology, metabolisms and ecology of participating microorganisms, 2) further investigation of the distribution of previously established processes, such as ANRA and DNRA, in marine ecosystems, 3) investigating the interactions between DNRA, denitrifier and anammox

communities in different marine environments, 4) exploring N cycle processes and the microbial communities involved in little-studied environments (e.g., $N_2$ fixation in the Indian Ocean and coastal zones, nitrification in coral reefs, and denitrification in deep sea environments), and 5) further analysis and modeling of the potential effects of anthropogenic activity on marine N processes, and how these effects interact with other biogeochemical processes.

*Author contributions.* Both authors contributed equally to the manuscript.

*Competing interests.* The authors declare that they have no conflict of interest.

*Acknowledgements.* R. Ramos's Master studies were supported by a CONACYT scholarship. Funding for this work was

granted by PAPIIT-UNAM No. IA201617. Financial support for language editing was provided by the Instituto de Ciencias del Mar y Limnología (UNAM).





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



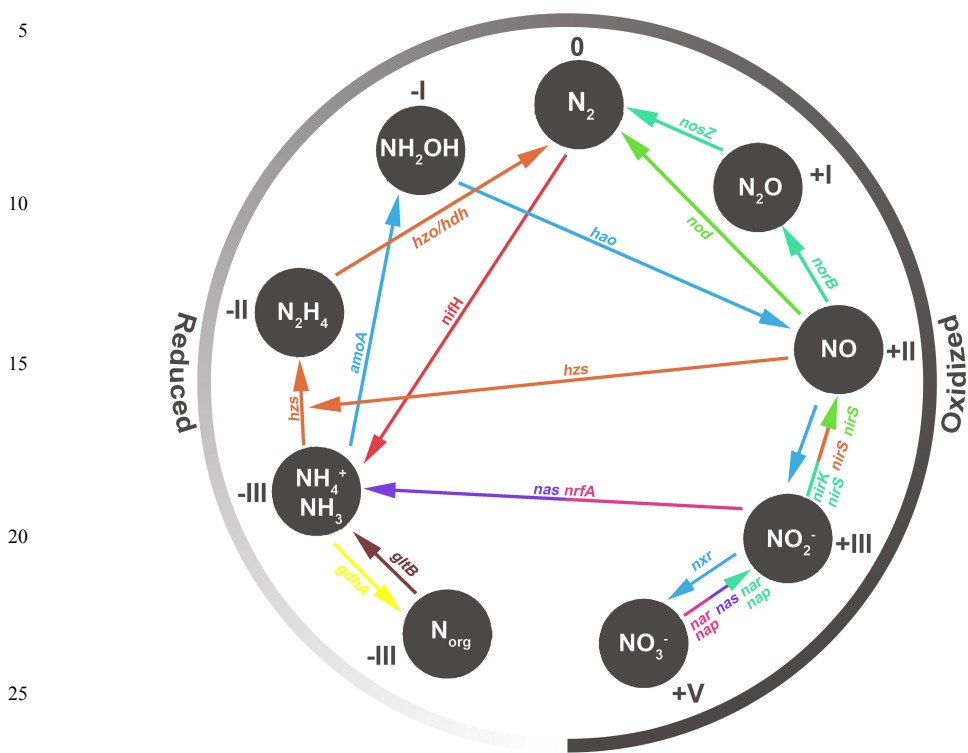

**Figure 1:** Nitrogen species involved in N cycling and its transformations. Each grey circle represents a N species, and the number next to each N species indicates its oxidation state. Colored arrows represent each N transformation and the marker genes involved: N₂ fixation (*nifH*) in red, nitrification (*amoA, hao, nxr*) in light blue, DNRA (*nar, nap, nrfA*) in magenta, ANRA (*nas*) in violet, denitrification (*nar, nap, nirK, nirS, norB, nosZ*) in emerald green, N-damo (*nod, nirS*) in green, anammox (*hzs, hdh*) in orange, N assimilation (*gdhA*) in yellow, and remineralization (*gltB*) in brown.



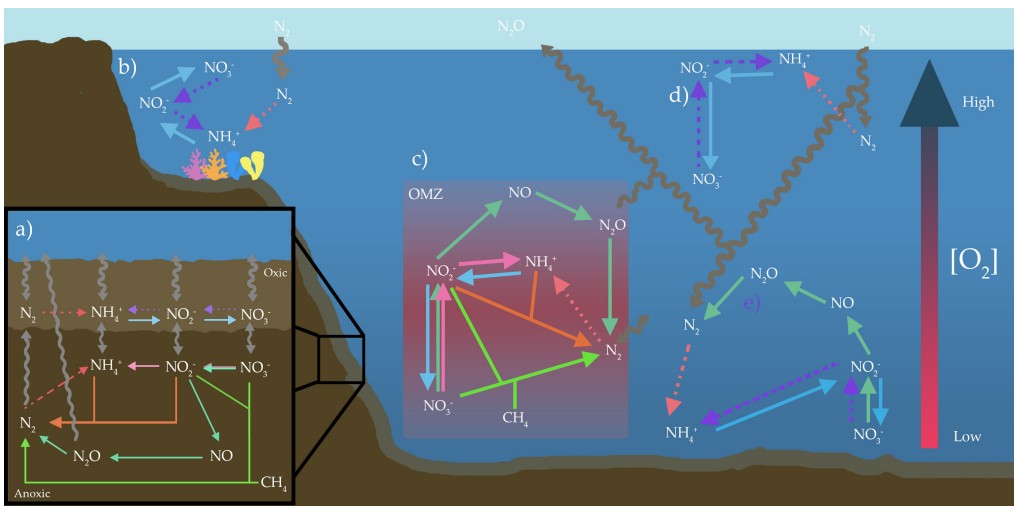

**Figure 2**. The main studied processes of the N cycle in different marine environments: a) coastal and deep ocean sediments, b) coral reefs,
10  c) oxygen minimum zones (OMZs), d) open ocean (with higher oxygen concentrations) and deep ocean (with lower oxygen
concentrations). Every colored arrow represents a N transformation: $N_2$ fixation (red), nitrification (light blue), DNRA (magenta), ANRA
(violet), denitrification (emerald green), N-damo (green), and anammox (orange). Continuous arrows represent dissimilatory processes,
while dashed arrows represent assimilatory processes. Grey curved arrows represent physical processes such as advection and diffusion.



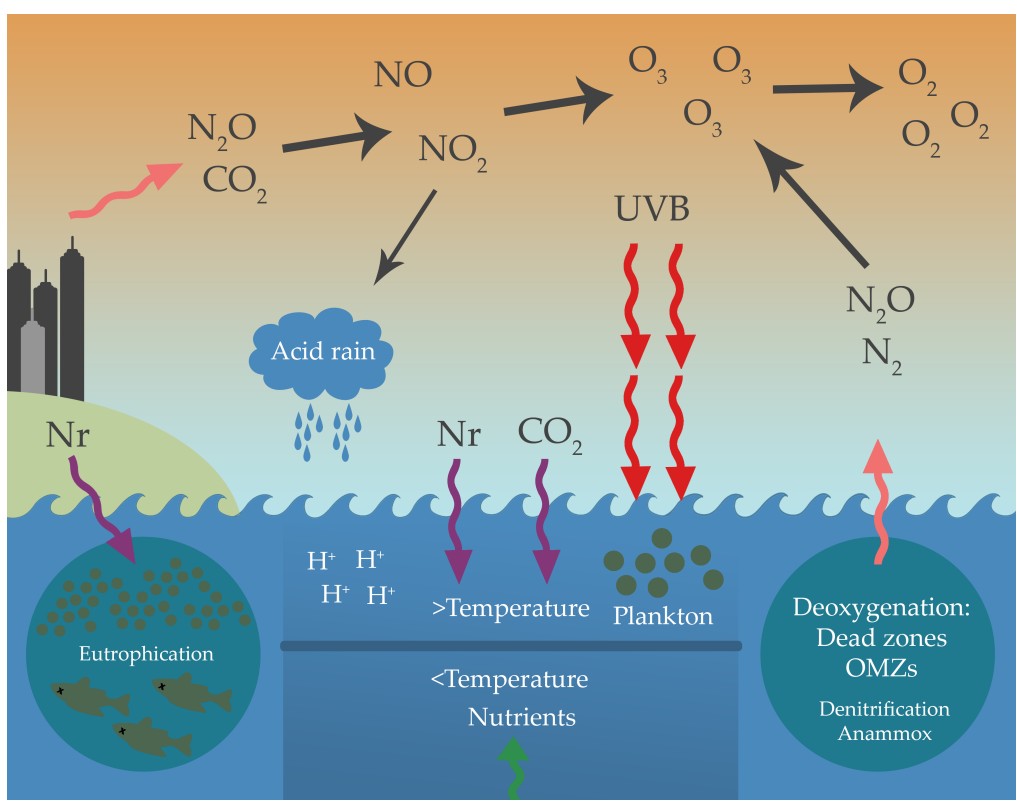

**Figure 3.** Anthropogenic activities and their effects on marine N cycling. Reactive N (Nr) is introduced into the marine ecosystems by runoff or atmospheric deposition, causing eutrophication, the formation of dead zones and the expansion of the ocean minimum zones (OMZs). The release of N oxides ($N_2O$, NO) from anthropogenic activities and oxygen-depleted zones causes stratospheric ozone depletion leading to higher UVB exposition, which produces the damage of marine life, acid rain and ocean warming. Ocean warming causes water stratification, deoxygenation, and the formation of dead zones. Dead zones and OMZs are hotspots for anammox and denitrification, causing N loss ($N_2$ and $N_2O$). Elevated atmospheric $CO_2$ acidifies seawater, decreasing pH-dependent N-cycling processes such as nitrification, and enhancing $N_2$ fixation.