# Peer review of "Reviews and syntheses: Processes and functional genes involved in nitrogen cycling in marine environments"

_Biogeosciences, 2018_

## Referee Comment (RC1) · B. Bayer (Referee) · 9 Aug 2018

The manuscript summarizes the very relevant topic of marine N cycling, addressing all major N cycling processes. Due to the existence of excellent reviews in this field (e.g. Voss et al 2013, Zehr and Kudela 2011, Lam and Kuypers 2011, Devol 2015), composing a review paper on this topic is a challenging task. In its current state, the manuscript does not represent a substantial contribution to the N cycle community. However, the authors mention a number of important aspects which could be elaborated further in order to focus more on new findings and ideas rather than repeating aspects that have already been reviewed. These aspects could potentially include (among others): more

emphasis on anthropogenic effects on the N cycle (and potentially connections with C cycle), budgets of N loss processes and factors favoring N-loss (e.g. denitrification vs. DNRA) and the production and consumption of climate-relevant gases such as N2O.

While the abstract provides a complete and concise summary of the topic, the overall representation of different aspects of the N cycle is not well-structured. Each N process is divided into sub-sections on factors affecting a specific process and on the distribution of organisms carrying out this process in the environment. However, as factors such as temperature, salinity, depth and oxygen concentrations define different environments, these two sections are highly repetitive and can be shortened and structured in a more concise way. In addition, some N process sections introduce other N processes (e.g. ANRA in the DNRA section). This issue could be overcome by a more detailed general introduction, which is at the moment rather short. A broader division into oxic vs anoxic, or nitrogen fixation/assimilation vs N loss processes could also help to guide the reader better through the different processes. In addition, some aspects on nitrifying microorganisms are not well represented in the manuscript. AOA are the dominant ammonia oxidizers in most parts of the global ocean, which does not become clear in the manuscript. Also, some statements regarding nitrifiers are too general e.g. it is not entirely clear why nitrifiers are mostly absent in surface waters and there are more potential explanations than UV light (specifically in the case of AOA). The whole process of nitrite oxidation is only mentioned in two sentences and the diversity and distribution of nitrite oxidizers is missing. While the recently discovered 'comammox' is a very interesting process and of great important for the N cycle community, thus far there is no evidence of the existence of this process in the marine environment. Hence, a whole section on comammox in a review paper on marine N cycling is not necessary.

---

## Referee Comment (RC2) · Anonymous Referee #2 · 10 Aug 2018

In their manuscript "Processes and functional genes involved in nitrogen cycling in marine environments," the authors have tried to assemble a comprehensive review of nitrogen cycling in the ocean. However, multiple recent and excellent reviews exist on this topic, foremost Kuypers et al. 2018, which is also frequently cited in this manuscript. Without wanting to offend the authors, it becomes quite clear when reading this manuscript that they cannot (yet) match the knowledge of the authors of some of these recent reviews. This makes me call the purpose of this manuscript into question. The apparent aim here is to present another review of all reactions and microorganisms involved in N cycling in marine systems, for which I must unfortunately say there is no need right now, and the authors lack the necessary expertise in many areas. This becomes apparent as many (recent) studies on different aspects of N cycling in the ocean are missing (e.g., Delmont et al. 2018, showing the abundance of non-cyanobacterial heterotrophic diazotrophs in marine metagenomes), and many pathways and proteins involved are incomplete and partly wrong (like for instance assimilatory nitrate and nitrite reduction). However, the authors do have and present a fairly good overview of the primer sets available and used for detecting the different steps of the N cycle. In view of this expertise, and the focus of the other available reviews, I would strongly advise the authors to focus this manuscript on a comprehensive review of the available tools to study the functional guilds involved in N cycling, which questions these can answer, and what are their limitations (which is the part I miss the most in the current manuscript). Here I would suggest to include discussions of limited coverage of some (most) primer sets and the existence of multiple pathways for the same reactions (e.g. in both assimilatory [nirA, nirBD, OTR/ONR] and dissimilatory [nirK, nirS, nrfAH] nitrite reduction). I would also advise the authors to be as complete on the processes they include into this review as possible. For instance, the nitrification section focuses almost exclusively on AOA and ignores the marine AOB (foremost Nitrosococcus) and especially NOB (Nitrospina and Nitrococcus, but also Nitrospira). On the other hand, comammox is included, even if not observed in marine systems so far. The same goes for N-DAMO, which discusses only the nitrite-dependent NC10 bacteria and only shortly mentions the nitrate-dependent archaea. If there is a lack of molecular tools to detect some of these groups (as I earlier advised this should be the focus), this should be stated and discussed. A review like this should then also include a critical discussion of the limitations of any PCR-based study, as many metagenomic-based studies have recently been published showing the amount of novelty that is missed by these approaches.

---

## Author Comment (AC1) · 31 Aug 2018

We would like to thank the Reviewer for the review and the valuable suggestions, which will help us to substantially improve our manuscript. We address each of your points raised (between quotation marks) below.

1. "The manuscript summarizes the very relevant topic of marine N cycling, addressing all major N cycling processes. Due to the existence of excellent reviews in this field (e.g. Voss et al 2013, Zehr and Kudela 2011, Lam and Kuypers 2011, Devol 2015), composing a review paper on this topic is a challenging task. In its current state, the manuscript does not represent a substantial contribution to the N cycle community. However, the

[Figure]

authors mention a number of important aspects which could be elaborated further in order to focus more on new findings and ideas rather than repeating aspects that have already been reviewed. These aspects could potentially include (among others): more emphasis on anthropogenic effects on the N cycle (and potentially connections with C cycle), budgets of N loss processes and factors favoring N-loss (e.g. denitrification vs. DNRA) and the production and consumption of climate-relevant gases such as N2O."

We share your opinion that writing a review paper on this topic is a very challenging task given the many great contributions made by other authors in recent years. However, the review papers you mention cover aspects of marine N cycling that differ from the main aim of our manuscript. For instance, Voss et al. (2013) only covered some of the major N processes (i.e. N2 fixation, anammox, and denitrification) in surface waters, OMZs and coastal environments with an anthropogenic perspective without delving into the participant microbial communities. In a similar way, Zehr and Kudela (2011) did an excellent review of the current understanding of marine N cycling, focusing in budgets and connections with the cycling of other nutrients and mentioning N assimilation, N2 fixation, nitrification, anammox, and denitrification without delving into the ecology and distribution of the microorganisms involved. Other recently published papers have covered the N cycling only in specific marine environments. For example, the review by Lam and Kuypers (2011) covered the major N cycling processes in OMZ or the review by Devol et al. (2015) focused on N2-producing processes occurring in marine sediments, such as anammox and denitrification. In our review we try to go further, summarizing the current knowledge on the N processes studied so far in many marine environments and including new findings and processes that previous reviews have not addressed, such as the recent findings of N-DAMO in OMZs and anoxic estuarine sediments, as well as the distribution of microorganisms involved in the N cycling in marine environments and the factors affecting it. Thus, we consider that with the proper modifications our review paper could represent a good contribution to the scientific community working in this field. We also agree that the manuscript can be improved and in the new revised document we will emphasize more on anthropogenic

effects on the N cycle and will include all the suggested recommendations.

2. "While the abstract provides a complete and concise summary of the topic, the overall representation of different aspects of the N cycle is not well-structured. Each N process is divided into sub-sections on factors affecting a specific process and on the distribution of organisms carrying out this process in the environment. However, as factors such as temperature, salinity, depth and oxygen concentrations define different environments, these two sections are highly repetitive and can be shortened and structured in a more concise way."

We think that the division into sub-sections for each N process reflects well what we try to show in our manuscript; however, we will revise it and try to restructure the information in order to make it more concise and less repetitive.

3. "In addition, some N process sections introduce other N processes (e.g. ANRA in the DNRA section). This issue could be overcome by a more detailed general introduction, which is at the moment rather short. A broader division into oxic vs anoxic, or nitrogen fixation/assimilation vs N loss processes could also help to guide the reader better through the different processes."

We appreciate a lot the Reviewer's suggestions. We agree that we can improve the general introduction giving more detail information, so we will further work on it and we will also divide the presentation of the nitrogen processes as the Reviewer recommends.

4. "In addition, some aspects on nitrifying microorganisms are not well represented in the manuscript. AOA are the dominant ammonia oxidizers in most parts of the global ocean, which does not become clear in the manuscript. Also, some statements regarding nitrifiers are too general e.g. it is not entirely clear why nitrifiers are mostly absent in surface waters and there are more potential explanations than UV light (specifically in the case of AOA). The whole process of nitrite oxidation is only mentioned in two sentences and the diversity and distribution of nitrite oxidizers is missing. While the

recently discovered 'comammox' is a very interesting process and of great important for the N cycle community, thus far there is no evidence of the existence of this process in the marine environment. Hence, a whole section on comammox in a review paper on marine N cycling is not necessary."

We thank again the Reviewer for the good recommendations. We will improve this section by expanding the information related to AOA, AOB and NOB communities; nevertheless, we cannot delve into the subject too much since our manuscript is intended to be a general review of the microbial communities involved in each of the different nitrogen processes studied in different marine environments. We agree with the Reviewer that there is no evidence of the existence of comammox in the ocean; then, we will remove the comammox section in the revised manuscript.

---

## Author Comment (AC2) · 31 Aug 2018

We would like to thank the Reviewer for the detailed suggestions. We have taken these comments into consideration and we are sure they will strongly help to improve our manuscript. We address each of the Reviewer's points raised below:

1. "In their manuscript "Processes and functional genes involved in nitrogen cycling in marine environments," the authors have tried to assemble a comprehensive review of nitrogen cycling in the ocean. However, multiple recent and excellent reviews exist on this topic, foremost Kuypers et al. 2018, which is also frequently cited in this manuscript. Without wanting to offend the authors, it becomes quite clear when reading

this manuscript that they cannot (yet) match the knowledge of the authors of some of these recent reviews. This makes me call the purpose of this manuscript into question. The apparent aim here is to present another review of all reactions and microorganisms involved in N cycling in marine systems, for which I must unfortunately say there is no need right now, and the authors lack the necessary expertise in many areas. This becomes apparent as many (recent) studies on different aspects of N cycling in the ocean are missing (e.g., Delmont et al. 2018, showing the abundance of non-cyanobacterial heterotrophic diazotrophs in marine metagenomes), and many pathways and proteins involved are incomplete and partly wrong (like for instance assimilatory nitrate and nitrite reduction)."

We thank the Reviewer for the comments. As we mentioned to Reviewer #1, we understand that writing a review paper on this topic is very risky given the great contributions made by other authors in recent years. However, none of these review papers have covered the main aspects we try to cover with our manuscript. For instance, the recent review of Kuypers et al. (2018) summarizes the current understanding of the microbial nitrogen-cycling network but does not focus on the microbially mediated N processes in marine ecosystems. Additionally, other recent papers reviewing N cycling processes in marine environments are too specific (e.g. Devol et al. in 2015 reviewing marine sediments or Lam and Kuypers in 2011 reviewing OMZs) or focus on other aspects of marine N cycling (e.g. Voss et al. in 2013 focusing on the anthropogenic effects or Zehr and Kudela in 2011 analysing the current understanding and identifying knowledge gaps). Again, while most of these papers cover biochemical, genetic or anthropogenic aspects of N cycling, in our manuscript we try to summarize the current knowledge on the nitrogen processes studied so far in the ocean, as well as the distribution of microorganisms involved in N cycling in marine environments and the factors affecting it, which are aspects that have not been covered in such an integral way so far. We are aware that there has been a great advance in the study of the marine nitrogen cycle in recent years and for that reason we believe it is appropriate to conduct a review in the field including the latest discoveries that have not been covered

in other review papers. We apologize for not including several recent studies; however, we have included over 50 studies from the last three years. In the case of the study by Delmont et al. 2018, we must say that our manuscript was completed months ago, and it is hard to keep up with every study published in the last months (in fact, Delmont et al. 2018 was published a few days after submitting our article to Biogeosciences). In the revised version we will include it, as well as others that have been published since we first sent the manuscript for revision. Additionally, we will correct and complete the metabolic pathways and proteins.

2. "However, the authors do have and present a fairly good overview of the primer sets available and used for detecting the different steps of the N cycle. In view of this expertise, and the focus of the other available reviews, I would strongly advise the authors to focus this manuscript on a comprehensive review of the available tools to study the functional guilds involved in N cycling, which questions these can answer, and what are their limitations (which is the part I miss the most in the current manuscript). Here I would suggest to include discussions of limited coverage of some (most) primer sets and the existence of multiple pathways for the same reactions (e.g. in both assimilatory [nirA, nirBD, OTR/ONR] and dissimilatory [nirK, nirS, nrfAH] nitrite reduction)."

We appreciate a lot the Reviewer's recommendations. We agree with the Reviewer that the tools used so far to study the functional guilds involved in marine N cycling, the limited coverage of most primer sets and the existence of multiple pathways for the same reactions are interesting topics for our review. Then, we will include these suggestions without deviating from the main purpose of our manuscript.

3. "I would also advise the authors to be as complete on the processes they include into this review as possible. For instance, the nitrification section focuses almost exclusively on AOA and ignores the marine AOB (foremost Nitrosococcus) and especially NOB (Nitrospina and Nitrococcus, but also Nitrospira). On the other hand, comammox is included, even if not observed in marine systems so far. The same goes for N-DAMO, which discusses only the nitrite-dependent NC10 bacteria and only shortly mentions

the nitrate-dependent archaea. If there is a lack of molecular tools to detect some of these groups (as I earlier advised this should be the focus), this should be stated and discussed. A review like this should then also include a critical discussion of the limitations of any PCR-based study, as many metagenomic-based studies have recently been published showing the amount of novelty that is missed by these approaches."

We appreciate the comments and agree with the criticisms made by the Reviewer. In the next version we will include the missing information on nitrifiers and n-damo. We will also remove the comammox section and include the reviewer's suggestions, again, without deviating from the main purpose of our manuscript.
* * *